# Need for self-medication using over-the-counter psychoactive agents: A national survey in Japan

**Akihiro Shiina**[1]*, **Tomihisa Niitsu**[2], **Masaomi Iyo**[2]

**1** Division of Medical Care and Treatment, Center for Forensic Mental Health, Chiba University, Chiba, Japan, **2** Department of Psychiatry, Graduate School of Medicine, Chiba University, Chiba, Japan

* shiina-akihiro@faculty.chiba-u.jp

## Abstract

Self-medication using over-the-counter (OTC) drugs is an option for the autonomous treatment of several health problems. However, the use of OTC drugs to treat psychiatric conditions remains controversial. To clarify opinions regarding the use of OTC drugs to treat psychiatric problems, we conducted an anonymous online survey of 3000 people in Japan. Participants were stratified into three groups according to their history of mental health problems. Few participants had engaged in self-medication using OTC drugs for psychiatric symptoms, with the exception of insomnia. Participants who had used OTC drugs reported feeling less satisfied with their experience compared with those who had consulted a specialist. Participants who had used sleeping pills were likely to hold relatively positive opinions regarding the use of OTC psychiatric drugs. In conclusion, the need for self-medication of psychiatric symptoms appears to be limited. Education and further research may be necessary to promote self-medication for proper treatment of psychiatric conditions in Japan.

## Introduction

Mental health is a major element of public healthcare [1]. The World Health Organization (WHO) has suggested that mental disorders are the greatest cause of decreased quality of life [2]. In many developed nations, the number of people with mental disorders is rapidly increasing [3]. Thus, mental health not only affects individuals, but also influences societal wellbeing [4].

Medications represent a core treatment strategy for health problems, including mental disorders. Among the many drugs produced by pharmaceutical companies, a growing number of psychoactive agents have been found to be effective in treating mental illnesses.

However, newly developed drugs can be expensive, along with consultations with medical specialists that are often needed to obtain a prescription. These factors can limit the accessibility of treatments in lower-income patients. Furthermore, although some governments and health insurance organizations work to mitigate medical costs, these endeavors are not always effective in making treatments sufficiently accessible.

Self-medication via the use of over-the-counter (OTC) drugs is a cost-effective alternative to consulting a specialist, particularly for citizens in regions with high medical costs. Self-

[igaku-rinri@office.chiba-u.jp](mailto:igaku-rinri@office.chiba-u.jp)) for researchers who meet the criteria for access to confidential data.

**Funding:** This work was supported by the public interest OTC drug self-medication promotion foundation (http://www.otc-spf.jp/) in Japan. We have no other conflicts of interest to report regarding this study.

**Competing interests:** This work was supported by the public interest OTC drug self-medication promotion foundation (http://www.otc-spf.jp/) in Japan. We have no other conflicts of interest to report regarding this study.

medication may be a particularly convenient and cost-effective treatment strategy for elderly people who are at an increased risk of illness [5].

Self-medication may include the use of psychoactive agents. For example, a considerable number of citizens in Hong Kong frequently use sleeping pills that are purchased as OTC drugs [6]. In the US, approximately 10% of citizens reported the use of OTC sleeping pills [7].

Self-medication via OTC drugs is considered by some to be a form of autonomy. In addition to enhanced convenience, OTC drugs may be advantageous compared with medical drugs prescribed by a physician in terms of confidentiality and ease of administration. Thus, in Japan and other developed nations, the utilization of OTC drugs is an important consideration for healthcare systems.

Compared with other developed nations, OTC drugs are not particularly popular in Japan. Historically, the Japanese national health insurance system has greatly influenced the medical industry and patient behavior. Furthermore, the cost of a consultation with a physician has continuously been relatively low in Japan. Therefore, self-medication culture has not been particularly widespread. In recent years, however, health insurance providers have been facing increasing fiscal challenges due to the rapidly aging society. To address this, the government introduced a tax exemption scheme for OTC drug utilization to encourage citizens to engage in self-treatment for non-serious illnesses.

In Japan, thousands of OTC drugs are available for purchase by any adult, including foreigners, without the guidance of a medical professional. The Japan Pharmaceutical Information Center classifies OTC drugs into 18 categories. In this system, sleep-aids, cold and fever medications, anti-drowsiness and anti-dizziness agents, and sedating agents for children are categorized as psychiatric drugs. Only one psychotropic compound (diphenhydramine) is permitted in sleeping pills. Antidepressants are not available as OTC drugs. However, St. John's Wart, which had been found to have antidepressant effects [8], is available as a dietary supplement as it is not categorized as a drug.

There are several issues to consider before encouraging individuals to adopt a self-medication strategy for the treatment of mental disorders. First, it may be difficult for patients to diagnose themselves and choose appropriate medications to treat specific types of mental illness. Second, some psychiatric drugs can lead to drug dependence [9]. For instance, a French survey revealed that a considerable number of patients abuse psychoactive OTC drugs [10].

Given these concerns, it is rational to restrict the personal usage of psychiatric drugs. However, there appears to be room for discussion regarding the provision of a self-help option to enable individuals to independently address mild psychiatric problems.

To summarize, there are several valid points supporting the introduction and promotion of self-medication for treating psychiatric problems. However, more data are needed regarding public needs and opinions with respect to the self-management of mental health using OTC medication.

To address this in the present study, we conducted a survey to examine opinions regarding self-medication using OTC compounds for treating psychiatric symptoms.

## Materials and methods

### Outline

This study was a cross-sectional exploratory study using an anonymous web-based questionnaire survey.

### Process

We arranged a contract with Rakuten Insight, a marketing industry in Japan, in September 2019. We delegated to Rakuten Insight several steps of the study process: the recruitment of

participants, construction of a web-based questionnaire and answer form, and data collection.

## Participants

The participants in this study were selected from a pool of individuals who had registered as study candidates on the Rakuten Insight website. Any persons willing to participate could enroll in the study on a first come, first served basis, unless they met the exclusion criteria. Any persons who were or who had relatives who were mental health professionals or employees of a pharmaceutical company were excluded from study participation.

## Stratification

At the beginning of the questionnaire, the participants were asked about their history of mental health problems. According to their answer to the first question, we stratified the participants into three groups: "patients" (those who were currently experiencing psychiatric problems at the time of the survey), "former-patients" (those who had previously experienced psychiatric problems but were no longer affected), and "non-patients" (those who had never experienced any psychiatric problems). We attempted to recruit 1000 participants in each group.

## Contents of the questionnaire

We presented several multiple-choice questions to the participants.

Representative psychiatric problems were categorized according to five symptoms: anxiety, depression, insomnia, hallucinations, and others.

We asked the participants whether they had experienced each symptom. We also asked whether the participants had taken specific actions, such as "consulting with a specialist," "consulting with a non-specialist," "taking OTC drugs," and "taking pharmaceutical drugs." Furthermore, we asked the participants to evaluate the efficiency of each solution they had attempted using a 5-grade Likert scale.

In another section of the questionnaire, we gathered the opinions of the participants regarding the efficacy and safety of OTC drugs for psychiatric use. These questions were also scored using a 5-grade Likert scale. Additionally, we presented positive and negative ideas regarding the psychiatric use of OTC drugs, and asked the participants whether they supported each idea.

In the last section, we collected demographic data from each participant.

## Statistical analyses

We statistically analyzed the gathered data using SPSS for Windows, version 24 (IBM Corp., Armonk, NY). We adopted the Pearson's chi-square ($\chi^2$) test, examination of homoscedasticity, an analysis of variance (ANOVA), the Games-Howell test, and the Mann–Whitney U-test according to the characteristics of each dataset. The level of significance was set at $P < 0.05$.

## Ethical issues

**General principles.** This study was an anonymous questionnaire survey of Japanese citizens. As the protocol included no intervention, there were no medical risks of participation.

We referred to the World Medical Association Declaration of Helsinki to confirm that this study did not violate the legislation. We also confirmed that this study did not violate any national rules for clinical research.

**Informed consent.**  Because this study was an anonymous questionnaire survey, we did not receive any personal information from the participants or from Rakuten Insight. We presented the purpose of the study on the front page of the questionnaire form. All respondents agreed to participate by sending in their answer form. Rakuten Insight compensated the participants for their involvement according to the contract.

**Applications.**  We submitted the protocol of this study in advance to the Ethics Committee of Chiba University Graduate School of Medicine, which approved the study (number 3555) on October 16th, 2019.

We registered this study in the UMIN Clinical Trials Registry, which is a national clinical research registration system, with the number R000044184UMIN000038754 on December 3rd, 2019.

**Funding and conflicts of interest.**  This work was supported by the public interest OTC drug self-medication promotion foundation (http://www.otc-spf.jp/) in Japan. We have no other conflicts of interest to report regarding this study.

## Results

### Response

Data collection was conducted between October 17th and October 21st, 2019. As planned, a total of 3,000 participants, including the patients group, former-patients group, and non-patients group (1000 participants in each group) completed the questionnaire.

### Demographic data

Representative data regarding the participant demographics are shown in Table 1. Homoscedasticity was denied in the three groups with P < 0.001 (Levene statistic = 19.331). The three

**Table 1. Participant demographic data.**

|  |  | Patients group | Former-patients group | Non-patients group | total |
|---|---|---|---|---|---|
| Age | Mean ± SD | 46.2±10.4 | 48±11.7 | 49.4±12.5 | 47.88±11.7 |
| Gander | Men | 530 (53.0%) | 551 (55.1%) | 590 (59.0%) | 1671 (55.7%) |
|  | Women | 470 (47.0%) | 449 (44.9%) | 410 (41.0%) | 1329 (44.3%) |
| Educational history | Junior high school | 47 (4.8%) | 20 (2.0%) | 17 (1.8%) | 84 (2.9%) |
|  | High school | 314 (32.0%) | 271 (27.6%) | 257 (26.9%) | 842 (28.8%) |
|  | Job training program | 206 (21.0%) | 215 (21.9%) | 227 (23.8%) | 648 (22.2%) |
|  | University | 414 (42.2%) | 477 (48.5%) | 454 (47.5%) | 1345 (46.1%) |
| Occupation | Employed | 359 (36.5%) | 447 (45.8%) | 429 (44.5%) | 1235 (42.4%) |
|  | Government official | 46 (4.7%) | 59 (6.0%) | 65 (6.7%) | 170 (5.8%) |
|  | Self-employed | 89 (9.1%) | 75 (7.7%) | 94 (9.7%) | 258 (8.8%) |
|  | Housekeeper | 117 (12.0%) | 121 (12.4%) | 127 (13.2%) | 365 (12.5%) |
|  | Part-time job | 163 (16.7%) | 150 (15.4%) | 126 (13.1%) | 439 (15.1%) |
|  | Student | 3 (0.3%) | 5 (0.5%) | 6 (0.6%) | 14 (0.5%) |
|  | Unemployed | 197 (20.2%) | 120 (12.3%) | 118 (12.2%) | 435 (14.9%) |
|  | No answer | 26 (N/A) | 23 (N/A) | 35 (N/A) | 84 (N/A) |
| Work schedule | Mainly daytime | 515 (56.1%) | 607 (64.5%) | 618 (67.9%) | 1740 (62.8%) |
|  | Mainly nighttime | 29 (3.2%) | 19 (2.0%) | 14 (1.5%) | 62 (2.2%) |
|  | Shift-work | 52 (5.7%) | 67 (7.1%) | 58 (6.4%) | 177 (6.4%) |
|  | No regular schedule | 322 (35.1%) | 248 (26.4%) | 220 (24.2%) | 790 (28.5%) |
|  | No answer | 82 (N/A) | 59 (N/A) | 90 (N/A) | 231 (N/A) |

*SD Standard Deviation, N/A not applicable (excluded from percentage calculation).

groups differed in terms of several demographic variables (sum of squares = 5140.105, degrees of freedom (df) = 2, mean square = 134.196, F = 19.151, P < 0.001). Specifically, the participants in the former-patients group were younger than those in the non-patients group (P < 0.05) but older than those in the patients group (P < 0.01). Regarding the gender of respondents, the proportion of women was relatively smaller in the non-patients group ($\chi^2$ = 7.514, df = 2, likelihood ratio = 7.529, P < 0.05). Regarding the educational history of the respondents, those in the patients group had a relatively lower level of education compared with those in the other two groups (P < 0.001). The patients group had a low employment rate ($\chi^2$ = 50.195, df = 12, likelihood ratio = 49.403, adjusted residual = 5.7, P < 0.001).

## Categories of mental health problems

In the patients group, 740 were suffering from anxiety, 780 from depression, 732 from insomnia, 84 from hallucinations, and 75 from other problems. In the former-patients group, 610 had suffered from anxiety, 764 from depression, 618 from insomnia, 39 from hallucinations, and 37 from other problems.

## Solutions for psychiatric problems

The proportions of each solution that participants selected to address their psychiatric problems are shown in Fig 1. Overall, consulting a specialist was the predominant solution, followed by taking pharmaceutical medication.

## Opinions of respondents

**The effectiveness of OTC drugs for psychiatric problems.** The opinions of the participants regarding the effectiveness of OTC drugs for the treatment of psychiatric problems are summarized in Table 2.

Comparing the two groups according to their experience of insomnia revealed that 1350 participants who had experienced insomnia had a more negative opinion about the effectiveness of OTC drugs for insomnia compared with 650 participants with no experience of

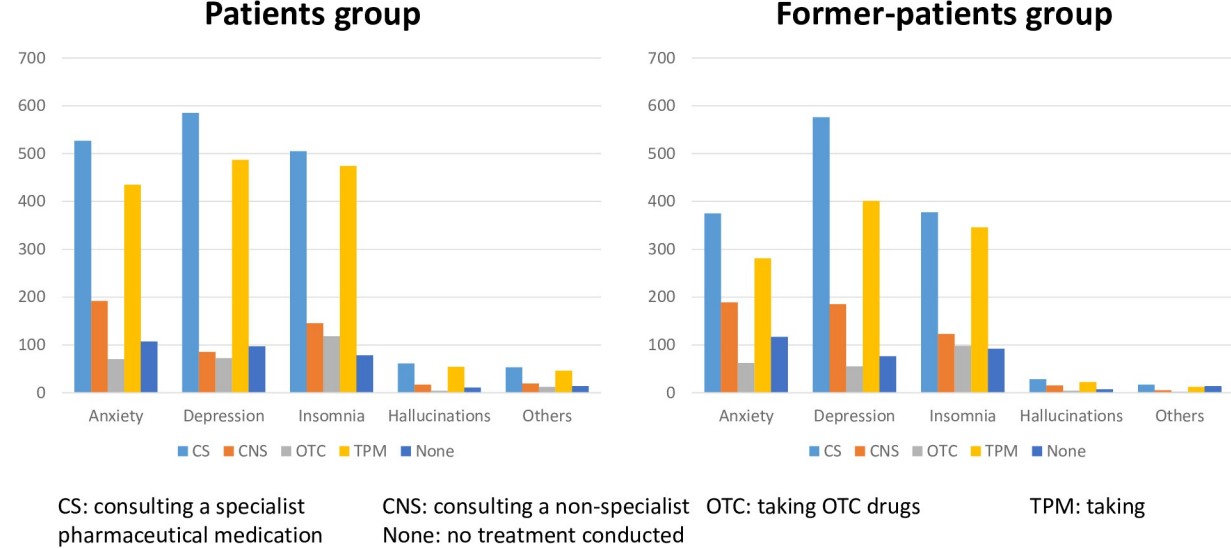

CS: consulting a specialist         CNS: consulting a non-specialist   OTC: taking OTC drugs          TPM: taking
pharmaceutical medication        None: no treatment conducted

**Fig 1. Solutions for each mental health symptoms.**

**Table 2. Participant opinions regarding the effectiveness of OTC drugs for the treatment of psychiatric problems.**

| | | Patients group | Former-patients group | Non-patients group | Total |
|---|---|---|---|---|---|
| Anxiety | Very effective | 24 (2.4%) | 27 (2.7%) | 22 (2.2%) | 73 (2.4%) |
| | Relatively effective | 91 (9.1%) | 102 (10.2%) | 83 (8.3%) | 276 (9.2%) |
| | Fairly effective | 426 (42.6%) | 472 (47.2%) | 531 (53.1%) | 1429 (47.6%) |
| | Relatively ineffective | 258 (25.8%) | 275 (27.5%) | 213 (21.3%) | 746 (24.9%) |
| | Completely ineffective | 201 (20.1%) | 124 (12.4%) | 151 (15.1%) | 476 (15.9%) |
| Depression | Very effective | 26 (2.6%) | 28 (2.8%) | 21 (2.1%) | 75 (2.5%) |
| | Relatively effective | 87 (8.7%) | 100 (10.0%) | 83 (8.3%) | 270 (9%) |
| | Fairly effective | 399 (39.9%) | 430 (43.0%) | 490 (49.0%) | 1319 (44.0%) |
| | Relatively ineffective | 251 (25.1%) | 276 (27.6%) | 226 (22.6%) | 753 (25.1%) |
| | Completely ineffective | 237 (23.7%) | 166 (16.6%) | 180 (18.0%) | 583 (19.4%) |
| Insomnia | Very effective | 35 (3.5%) | 43 (4.3%) | 38 (3.8%) | 116 (3.9%) |
| | Relatively effective | 163 (16.3%) | 222 (22.2%) | 224 (22.4%) | 609 (20.3%) |
| | Fairly effective | 442 (44.2%) | 488 (48.8%) | 522 (52.2%) | 1452 (48.4%) |
| | Relatively ineffective | 195 (19.5%) | 155 (15.5%) | 123 (12.3%) | 473 (15.8%) |
| | Completely ineffective | 165 (16.5%) | 92 (9.2%) | 93 (9.3%) | 350 (11.7%) |
| Hallucination | Very effective | 16 (1.6%) | 21 (2.1%) | 22 (2.2%) | 59 (2.0%) |
| | Relatively effective | 47 (4.7%) | 59 (5.9%) | 57 (5.7%) | 163 (5.4%) |
| | Fairly effective | 449 (44.9%) | 506 (50.6%) | 477 (47.7%) | 1432 (47.7%) |
| | Relatively ineffective | 197 (19.7%) | 205 (20.5%) | 221 (22.1%) | 623 (20.8%) |
| | Completely ineffective | 291 (29.1%) | 209 (20.9%) | 223 (22.3%) | 723 (24.1%) |
| Other psychiatric problems | Very effective | 16 (1.6%) | 17 (1.7%) | 16 (1.6%) | 49 (1.6%) |
| | Relatively effective | 38 (3.8%) | 39 (3.9%) | 20 (2.0%) | 97 (3.2%) |
| | Fairly effective | 576 (57.6%) | 670 (67.0%) | 706 (70.6%) | 1952 (65.1%) |
| | Relatively ineffective | 168 (16.8%) | 146 (14.6%) | 112 (11.2%) | 426 (14.2%) |
| | Completely ineffective | 202 (20.2%) | 128 (12.8%) | 146 (14.6%) | 476 (15.9%) |

insomnia (P < 0.001). However, when the data were stratified according to the experience of using OTC drugs for insomnia, no statistically significant differences were detected between the opinions of 216 participants who had used OTC drugs for insomnia and those of 1134 participants who had not (Fig 2).

**The safety of using OTC drugs to treat psychiatric problems.** The opinions of the participants regarding the safety of using OTC drugs to treat psychiatric problems are summarized in Table 3.

When we compared the two groups according to their experience of insomnia, no statistically significant differences were detected between the opinions of the 1350 participants who had experienced insomnia and those of the 650 participants with no experience of insomnia. However, 216 participants who had experienced insomnia had a more positive opinion about the safety of using OTC drugs to treat insomnia compared with 1134 participants who had not previously used OTC drugs for insomnia (P < 0.001) (Fig 3).

**Advantages of OTC drugs for psychiatric use.** We asked the participants a question regarding the general advantages of OTC drugs for psychiatric use. In this section, we presented statements that reflected particular beliefs and asked the participants to indicate whether they agreed with each statement.

The results regarding the advantages of OTC drugs for psychiatric use are shown in Fig 4. A total of 295 participants in the patients group, 304 in the former-patients group, and 262 in the non-patients group agreed with the idea that OTC drugs were "affordable." A total of 259

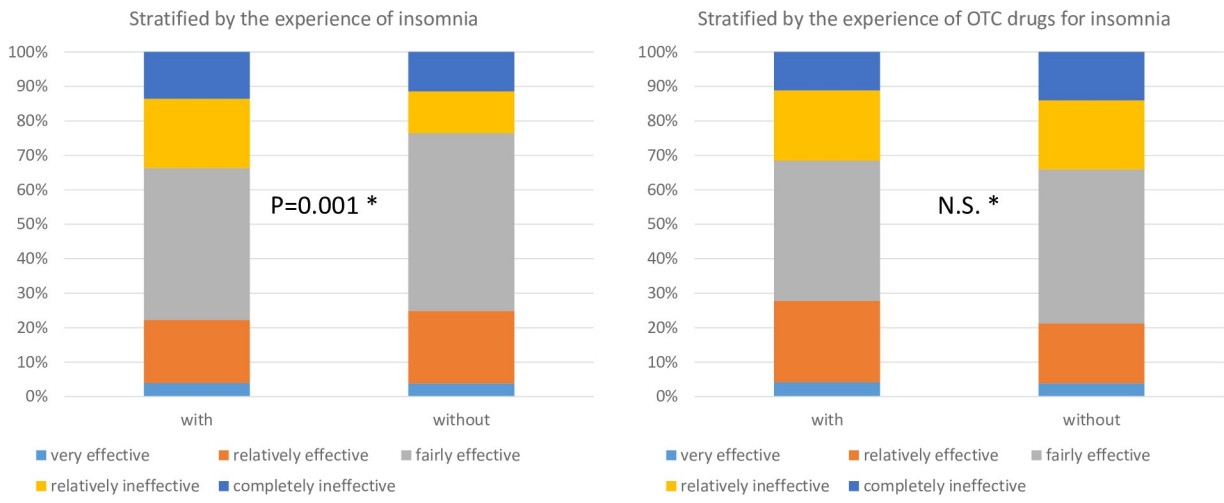

* Mann-Whitney-U test

**Fig 2. Opinion regarding the effectiveness of OTC drugs for insomnia.**

participants in the patients group, 294 in the former-patients group, and 213 in the non-patients group agreed with the idea that OTC drugs were "flexible." A total of 369 participants in the patients group, 406 in the former-patients group, and 327 in the non-patients group agreed with the idea that OTC drugs are "private." A total of 169 participants in the patients group, 145 in the former-patients group, and 117 in the non-patients group agreed with the idea that OTC drugs are "safe." A total of 349 participants in the patients group, 286 in the former-patients group, and 394 in the non-patients group did not agree with any of the statements for that question. Those in the former-patients group were more likely to consider OTC drugs for psychiatric use to be flexible ($\chi^2$ = 17.359, df = 2, P < 0.001, adjusted residual = 3.4) and private ($\chi^2$ = 13.445, df = 2, P < 0.01, adjusted residual = 3.1) compared with those in the non-patients group (adjusted residual = −3.8, −3.2, respectively). Additionally, the participants in the patients group were more likely to consider OTC drugs to be safe for psychiatric use ($\chi^2$ = 11.011, df = 2, P < 0.01, adjusted residual = 2.8) compared with those in the non-patients group (adjusted residual = −2.9).

**Disadvantages of OTC drugs for psychiatric use.** The results regarding the disadvantages of OTC drugs are shown in Fig 5. A total of 652 participants in the patients group, 647 in the former-patients group, and 523 in the non-patients group agreed with the idea that it is "difficult" for individuals to make precise decisions about the treatment of psychiatric problems without medical advice. A total of 540 participants in the patients group, 529 in the former-patients group, and 413 in the non-patients group agreed with the idea that OTC drugs carry a risk of "dependence." A total of 180 participants in the patients group, 191 in the former-patients group, and 188 in the non-patients group agreed that they were "reluctant" regarding the use of OTC drugs. A total of 449 participants in the patients group, 390 in the former-patients group, and 350 in the non-patients group agreed with the idea that OTC drugs are "risky." A total of 121 participants in the patients group, 108 in the former-patients group, and 257 in the non-patients group did not agree with any ideas presented in the question. The participants in the non-patients group tended not to consider the use of OTC drugs for treating psychiatric problems to be difficult ($\chi^2$ = 48.787, df = 2, P < 0.001, adjusted residual = −6.7) or to carry a risk of dependence ($\chi^2$ = 39.614, df = 2, P < 0.01, adjusted residual =

**Table 3. Participant opinions regarding the safety of OTC drugs for the treatment of psychiatric problems.**

| | | Patients group | Former-patients group | Non-patients group | Total |
|---|---|---|---|---|---|
| Anxiety | Very safe | 18 (1.8%) | 20 (2.0%) | 21 (2.1%) | 59 (2.0%) |
| | Relatively safe | 81 (8.1%) | 82 (8.2%) | 76 (7.6%) | 239 (8.0%) |
| | Fairly safe | 522 (52.2%) | 551 (55.1%) | 572 (57.2%) | 1645 (54.8%) |
| | Relatively unsafe | 231 (23.1%) | 250 (25%) | 198 (19.8%) | 679 (22.6%) |
| | Completely unsafe | 148 (14.8%) | 97 (9.7%) | 133 (13.3%) | 378 (12.6%) |
| Depression | Very safe | 16 (1.6%) | 23 (2.3%) | 21 (2.1%) | 60 (2.0%) |
| | Relatively safe | 82 (8.2%) | 64 (6.4%) | 55 (5.5%) | 201 (6.7%) |
| | Fairly safe | 492 (49.2%) | 519 (51.9%) | 533 (53.3%) | 1544 (51.5%) |
| | Relatively unsafe | 237 (23.7%) | 265 (26.5%) | 223 (22.3%) | 725 (24.2%) |
| | Completely unsafe | 173 (17.3%) | 129 (12.9%) | 168 (16.8%) | 470 (15.7%) |
| Insomnia | Very safe | 21 (2.1%) | 25 (2.5%) | 26 (2.6%) | 72 (2.4%) |
| | Relatively safe | 114 (11.4%) | 126 (12.6%) | 131 (13.1%) | 371 (12.4%) |
| | Fairly safe | 519 (51.9%) | 556 (55.6%) | 548 (54.8%) | 1623 (54.1%) |
| | Relatively unsafe | 209 (20.9%) | 208 (20.8%) | 189 (18.9%) | 606 (20.2%) |
| | Completely unsafe | 137 (13.7%) | 85 (8.5%) | 106 (10.6%) | 328 (10.9%) |
| Hallucination | Very safe | 17 (1.7%) | 19 (1.9%) | 20 (2.0%) | 56 (1.9%) |
| | Relatively safe | 59 (5.9%) | 48 (4.8%) | 44 (4.4%) | 151 (5.0%) |
| | Fairly safe | 479 (47.9%) | 524 (52.4%) | 510 (51.0%) | 1513 (50.4%) |
| | Relatively unsafe | 231 (23.1%) | 240 (24.0%) | 219 (21.9%) | 690 (23.0%) |
| | Completely unsafe | 214 (21.4%) | 169 (16.9%) | 207 (20.7%) | 590 (20.0%) |
| Other psychiatric problems | Very safe | 16 (16%) | 20 (2.0%) | 18 (1.8%) | 54 (1.8%) |
| | Relatively safe | 56 (5.6%) | 43 (4.3%) | 28 (2.8%) | 127 (4.2%) |
| | Fairly safe | 586 (58.6%) | 641 (64.1%) | 660 (66.0%) | 1887 (62.9%) |
| | Relatively unsafe | 183 (18.3%) | 187 (18.7%) | 156 (15.6%) | 526 (17.5%) |
| | Completely unsafe | 159 (15.9%) | 109 (10.9%) | 138 (13.8%) | 406 (13.5%) |

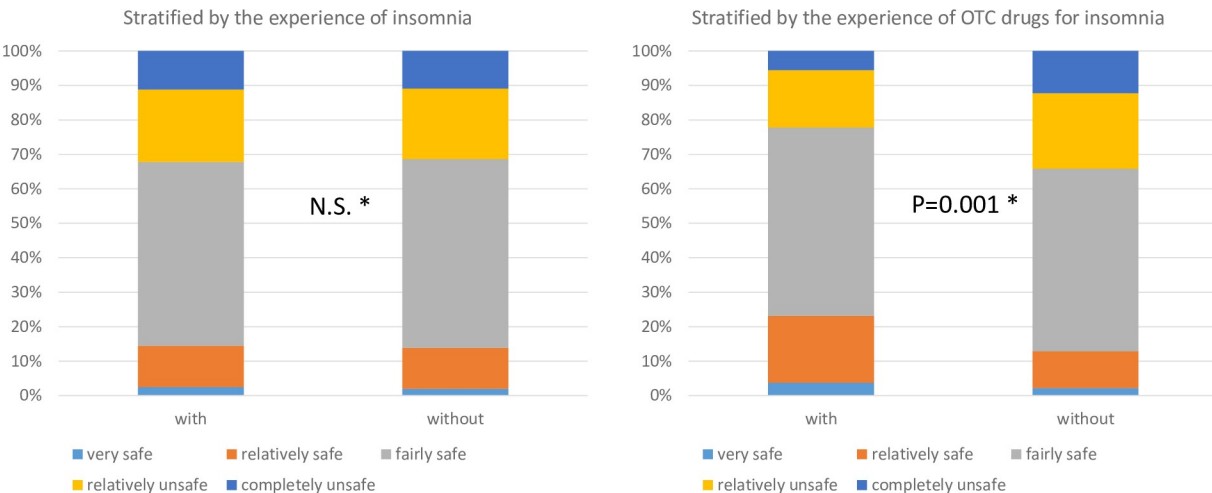

* Mann-Whitney-U test

**Fig 3. Opinion regarding the safety of OTC drugs for insomnia.**

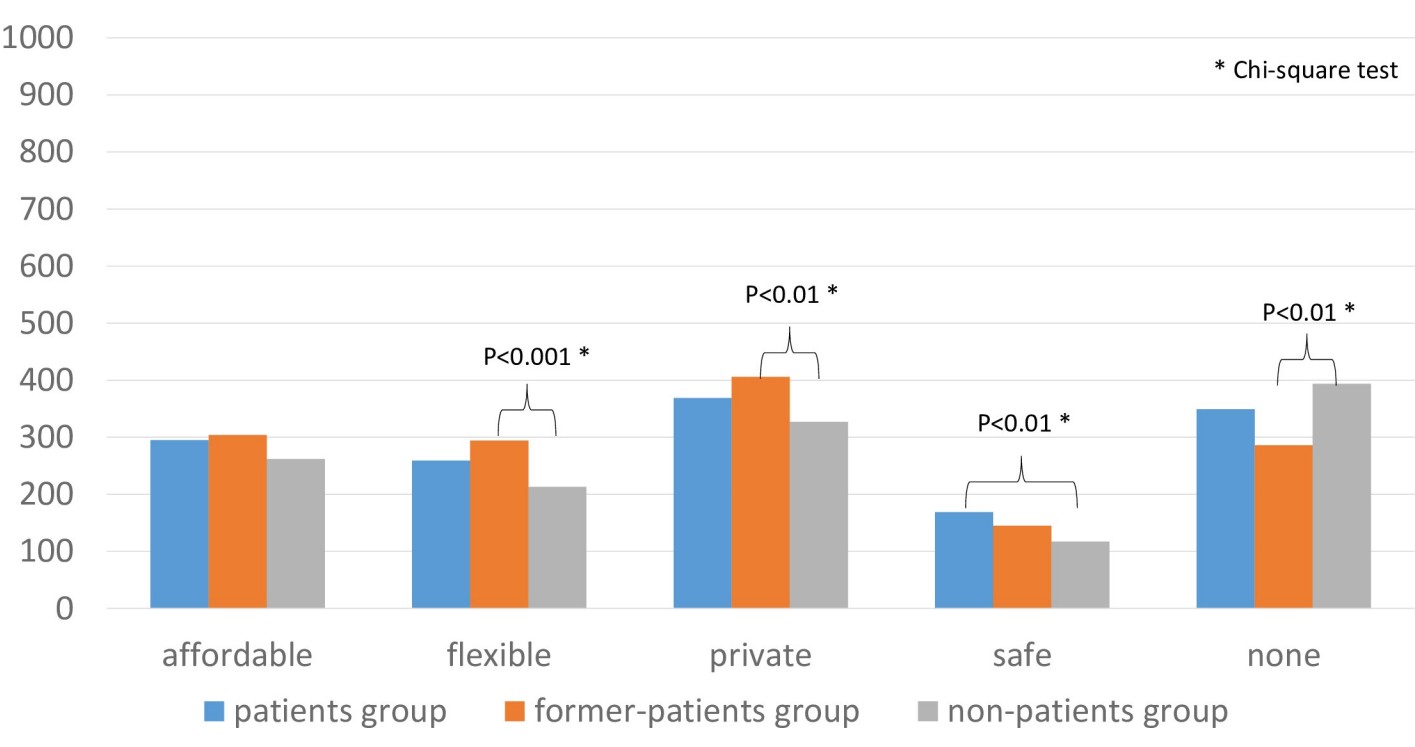

**Fig 4. Advantages of OTC drugs for psychiatric use.**

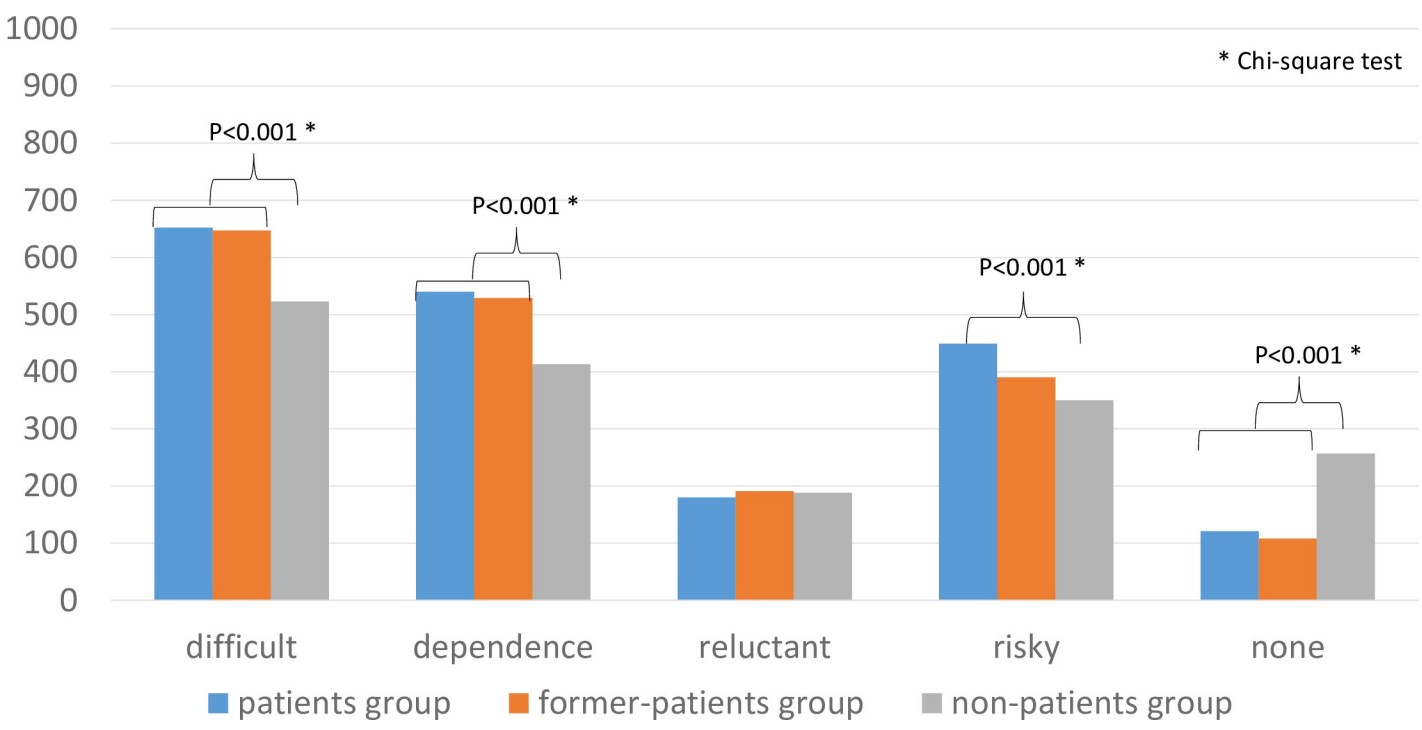

**Fig 5. Disadvantages of OTC drugs for psychiatric use.**

−6.3) compared with those in the other two groups. Also, participants in the non-patients group were less likely to consider OTC drugs for psychiatric use as risky ($\chi^2$ = 20.734, df = 2, P < 0.001, adjusted residual = −3.7) compared with those in the patients group (adjusted residual = 4.2).

## Discussion

In the present study, we conducted a survey to obtain information about the experiences and opinions of Japanese citizens regarding the use of OTC drugs for treating psychiatric conditions. A total of 3000 participants completed the questionnaire. To our knowledge, few reports have examined public experience and opinions regarding the use of OTC drugs for treating psychiatric problems. In a survey of the general population in the UK, 3.5% of citizens reported taking psychotropic medications, and of these, 7.2% took OTC psychiatric drugs [11]. However, the researchers did not ask the participants about the subjective effectiveness of the drugs. In the Netherlands, 1.7% of the participants in a survey study reported taking OTC psychiatric drugs, and 33.3% of this group also took psychiatric medications that had been prescribed by a medical professional. The report found that OTC psychiatric drug users were more likely to be female, have psychological problems, and to be recipients of social support, compared with those who did not take OTC medications [12]. Differences in the study design impede a direct comparison of the results with those of the present study.

Our results indicate that women with low levels of education and unstable employment are more likely to experience psychiatric problems, which is consistent with previous research [13]. However, we cannot make any conclusions regarding causal relationships between these factors because our study was cross-sectional.

Many of the study participants who reported experiencing psychiatric problems, especially those with anxiety and depression, indicated that they had consulted a medical professional. Relatively few of these participants had utilized OTC drugs. This is likely because there are no OTC drugs available for treating anxiety or depression in Japan, and the Japanese government offers national health insurance to all Japanese citizens. Furthermore, relatively few participants answered that they had experienced hallucinations. This is in contrast with a report indicating that 15% of psychologically healthy students have had at least one experience of hallucination [14]. Thus, it is possible that the prevalence rate of hallucinations may have been underrepresented in the present study, or that the participant group was healthier than the general population.

Our data indicate that, generally, the participants did not think that OTC drugs were appropriate for treating psychiatric conditions. For example, participants who had experienced insomnia were more likely to distrust the effectiveness of OTC insomnia drugs. However, participants who had used OTC drugs for insomnia did not appear to be particularly disappointed with the drug efficacy. It appears that although such medications are available, few participants chose to use OTC drugs to treat insomnia.

Regarding safety, although some participants believed that OTC sleeping pills were risky, this attitude was less prevalent in participants who had experience taking sleeping pills.

Our findings suggest that more education regarding the use of OTC sleeping pills is needed to encourage the use of OTC medication in Japan. Some individuals may be hesitant to use psychiatric drugs, regardless of the drug characteristics. However, education regarding the risks and benefits of OTC drugs may influence behavior in this group. Indeed, we found that those with prior experiences of psychiatric problems were more likely to encourage the use of OTC drugs, which indicates that they may have benefited from their use in the past.

Given their frontline position in interacting with people who may choose to undergo self-medication, the role of pharmacists should be emphasized in promoting the proper use of OTC drugs. Like those in the general population, many pharmacists in Japan may underestimate the effectiveness of OTC sleeping pills [15]. If pharmacists encourage clients with minimal symptoms to use OTC drugs, while recommending that patients seek a consultation with a psychiatrist for serious medical problems, may lead to an increase in the beneficial use of OTC drugs.

Positive use of OTC drugs may also increase if medical doctors are effectively educated regarding the benefits of OTC drugs. In Japan, very few studies have examined the perceptions of physicians regarding OTC drugs. Considering that some researchers have reported on the risk of dependency of OTC drugs [16], it is necessary to encourage physicians to become more familiar with OTC drugs, especially those that carry a risk of dependency. In addition, collaboration between pharmacists and physicians will likely be helpful in providing therapeutic alternatives to patients when appropriate.

Healthy foods and supplements should also be points of discussion regarding the promotion of mental health using self-medication strategies. For instance, several compounds such as omega-3 fatty acids [17] and vitamin D [18] are considered to mitigate psychiatric symptoms. In Japan, evidence regarding the current usage of nutritional supplements is scarce. Scientific data are needed concerning how often people take supplements and the degree to which they are beneficial for mental health. Conversely, many cases of caffeine poisoning due to the excessive consumption of tablets and/or energy drinks have been reported [19]. Appropriate usage of these materials should be established.

The present study has several limitations. First, we used a marketing company to conduct this web-based survey, leading to the possibility of sampling bias. Only people with internet access who had registered with the marketing service participated in this study. Thus, the degree to which the study findings are representative to the general public may be limited. Second, we classified the participants into three groups. Therefore, the percentages of participants who chose each answer to the questions in the survey are not likely to reflect the proportion of the general population who hold each opinion. Third, we only gathered subjective responses from each participant. We did not use any qualified rating scales to evaluate the participant responses, and did not have a way to verify the participant reports via medical records. This further limits the generalizability of the present data.

## Conclusion

Our data indicate that reliance on OTC drugs for psychiatric use is limited in Japanese citizens. However, participants who used sleeping pills had more positive opinions about self-medication for psychiatric problems and may be less reluctant to use medication to address mental health problems in the future. Further research and education are required to encourage self-medication of psychiatric symptoms using OTC drugs.

## Supporting information

**S1 File. Items included in the online questionnaire (translated from Japanese to English).** (DOCX)

## Author Contributions

**Conceptualization:** Akihiro Shiina, Tomihisa Niitsu.

**Data curation:** Akihiro Shiina.

**Formal analysis:** Akihiro Shiina.

**Funding acquisition:** Akihiro Shiina.

**Investigation:** Akihiro Shiina.

**Methodology:** Akihiro Shiina, Tomihisa Niitsu.

**Project administration:** Akihiro Shiina, Masaomi Iyo.

**Resources:** Akihiro Shiina.

**Supervision:** Masaomi Iyo.

**Validation:** Akihiro Shiina, Tomihisa Niitsu.

**Visualization:** Akihiro Shiina, Tomihisa Niitsu.

**Writing – original draft:** Akihiro Shiina.

**Writing – review & editing:** Tomihisa Niitsu.

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
