## [Decision Letter · Decision Letter 0]

3 Nov 2020

PONE-D-20-31979

Need for self-medication using over-the-counter psychoactive agents: A national survey in Japan

PLOS ONE

Dear Dr. Shiina,

Thank you for submitting your manuscript to PLOS ONE. After careful consideration, we feel that it has merit but does not fully meet PLOS ONE’s publication criteria as it currently stands. Therefore, we invite you to submit a revised version of the manuscript that addresses the points raised during the review process.

I would like you to address all of the reviewers' comments.

We look forward to receiving your revised manuscript.

Kind regards,

Kazutaka Ikeda, Ph.D.

Academic Editor

PLOS ONE

Journal Requirements:

3.We note that you have indicated that data from this study are available upon request. PLOS only allows data to be available upon request if there are legal or ethical restrictions on sharing data publicly. For information on unacceptable data access restrictions, please see http://journals.plos.org/plosone/s/data-availability#loc-unacceptable-data-access-restrictions.

4.Thank you for stating the following in the Competing Interests section:

[This work was supported by the public interest OTC drug self-medication promotion foundation (http://www.otc-spf.jp/) in Japan. We have no other conflicts of interest to report regarding this study.].

Reviewers' comments:

Reviewer's Responses to Questions

**Comments to the Author**

1. Is the manuscript technically sound, and do the data support the conclusions?

Reviewer #1: Partly

Reviewer #2: Yes

2. Has the statistical analysis been performed appropriately and rigorously? 

Reviewer #1: Yes

Reviewer #2: Yes

3. Have the authors made all data underlying the findings in their manuscript fully available?

Reviewer #1: Yes

Reviewer #2: Yes

4. Is the manuscript presented in an intelligible fashion and written in standard English?

Reviewer #1: No

Reviewer #2: Yes

5. Review Comments to the Author

Reviewer #1: This is a well-designed study with a novel objective.

However, I would like the authors to address the following points.

- Major points

I'm wondering OTC drugs available in Japan. The authors may want to show the kinds of OTC drugs and their potential indications for mental conditions.

The authors may want to compare their results with previous papers from different countries to discuss the novelty, strengths, consistency, and limitations of this study.

Please add the limitation of subjective ratings to the discussion section.

I'm wondering why they conclude that Japanese people are lacking in information on OTC drugs for psychiatric problems based on the results of this study.

- Minor points

Please describe numbers as mean and SD for continuous variables or numbers and % for categorical variables in the tables.

Please align the number of decimal places.

Please ask for a native English check.

Reviewer #2: The paper entitled “Need for self-medication using over-the-counter psychoactive agents: A national survey in Japan” is a well-controlled study and the authors provide new and useful information. Overall I found this study to be interesting and the first study to examine public opinion regarding the use of OTC drugs to treat psychiatric symptoms. However, there are some issues that the authors should address.

1. The authors described that “Further research and education are required to encourage self-medication of psychiatric symptoms using OTC drugs” in conclusion. The authors should give an example and explain in detail for the pharmacists especially. Please also discuss the relationship between doctors and pharmacists.

2. In Overseas, many health foods and/or supplements are widespread, and then widely used. In Japan, it seems that their amount used and kinds are less than those in oversea. The author should discuss not only OTC but also health foods and supplements as self-medication. How much they are used as self-medication for psychoactive agents. There are some health foods and supplements used for symptoms.

6. PLOS authors have the option to publish the peer review history of their article (what does this mean?). If published, this will include your full peer review and any attached files.

Reviewer #1: No

Reviewer #2: No

---

## [Author Response · Author response to Decision Letter 0]

22 Dec 2020

Response to the reviewers

Reviewer #1: This is a well-designed study with a novel objective.

However, I would like the authors to address the following points.

- Major points

I'm wondering OTC drugs available in Japan. The authors may want to show the kinds of OTC drugs and their potential indications for mental conditions.

As described in the introduction section, many OTC drugs are available in Japan. But only sleeping pills are categorized as psychiatric OTC drugs. We have added some information about the current availability of OTC drugs in Japan as per the reviewer’s comment.

The authors may want to compare their results with previous papers from different countries to discuss the novelty, strengths, consistency, and limitations of this study.

Thanks for your advice. When we searched for similar studies conducted previously, we found that very few reports have examined self-medication using OTC drugs in psychiatry. We cited some relevant studies as suggested and included further information about this issue in the discussion section.

Please add the limitation of subjective ratings to the discussion section.

Thank you for your suggestion. We have added this as a study limitation.

I'm wondering why they conclude that Japanese people are lacking in information on OTC drugs for psychiatric problems based on the results of this study.

We speculate that people who have experienced psychiatric problems are more likely to encourage the use of OTC drugs because of their experience with such pharmaceuticals and receipt of information from medical practitioners. However, we deleted this sentence from the conclusion since it may be confusing to readers. Thank you for your suggestion.

- Minor points

Please describe numbers as mean and SD for continuous variables or numbers and % for categorical variables in the tables.

Please align the number of decimal places.

Thank you for your suggestion. We have amended the text in each table.

Please ask for a native English check.

Thank you for your suggestion. The original manuscript was edited by a proofreading service, but we understand that some confusing phrases may have remained in the text. Before submitting the revised edition, we had the manuscript rechecked by a native English editor.

Reviewer #2: The paper entitled “Need for self-medication using over-the-counter psychoactive agents: A national survey in Japan” is a well-controlled study and the authors provide new and useful information. Overall I found this study to be interesting and the first study to examine public opinion regarding the use of OTC drugs to treat psychiatric symptoms. However, there are some issues that the authors should address.

1. The authors described that “Further research and education are required to encourage self-medication of psychiatric symptoms using OTC drugs” in conclusion. The authors should give an example and explain in detail for the pharmacists especially. Please also discuss the relationship between doctors and pharmacists.

Thank you for your suggestion. We added some information about the importance of education regarding OTC drugs and factors related to the promotion of OTC drugs, including the importance of collaborations between physicians and pharmacists, in the discussion section. However, few previous studies have examined this issue in Japan, so we were unable to give specific examples.

2. In Overseas, many health foods and/or supplements are widespread, and then widely used. In Japan, it seems that their amount used and kinds are less than those in oversea. The author should discuss not only OTC but also health foods and supplements as self-medication. How much they are used as self-medication for psychoactive agents. There are some health foods and supplements used for symptoms.

Thank you for your suggestion. We are aware that OTC supplements may have an important role in the issue of self-medication. Unfortunately, we did not gather data regarding supplements in the present study. To our knowledge, few studies have examined this issue in Japan. In the last part of the discussion section, we added some information regarding relevant ideas for further studies.

---

## [Decision Letter · Decision Letter 1]

11 Jan 2021

Need for self-medication using over-the-counter psychoactive agents: A national survey in Japan

PONE-D-20-31979R1

Dear Dr. Shiina,

We’re pleased to inform you that your manuscript has been judged scientifically suitable for publication and will be formally accepted for publication once it meets all outstanding technical requirements.

Kind regards,

Kazutaka Ikeda, Ph.D.

Academic Editor

PLOS ONE

Additional Editor Comments (optional):

Reviewers' comments:

Reviewer's Responses to Questions

**Comments to the Author**

1. If the authors have adequately addressed your comments raised in a previous round of review and you feel that this manuscript is now acceptable for publication, you may indicate that here to bypass the “Comments to the Author” section, enter your conflict of interest statement in the “Confidential to Editor” section, and submit your "Accept" recommendation.

Reviewer #1: All comments have been addressed

Reviewer #2: All comments have been addressed

2. Is the manuscript technically sound, and do the data support the conclusions?

Reviewer #1: Yes

Reviewer #2: Yes

3. Has the statistical analysis been performed appropriately and rigorously? 

Reviewer #1: Yes

Reviewer #2: Yes

4. Have the authors made all data underlying the findings in their manuscript fully available?

Reviewer #1: Yes

Reviewer #2: Yes

5. Is the manuscript presented in an intelligible fashion and written in standard English?

Reviewer #1: Yes

Reviewer #2: Yes

6. Review Comments to the Author

Reviewer #1: They addressed my comments in the previous review. I’d agree to the acceptance of this manuscript. Of another note, they should have detailed the revision content in the letter.

Reviewer #2: This manuscript was improved over the original version, and some questions seem to have been adequately addressed.

7. PLOS authors have the option to publish the peer review history of their article (what does this mean?). If published, this will include your full peer review and any attached files.

Reviewer #1: No

Reviewer #2: No

---

## [Editor Report · Acceptance letter]

15 Jan 2021

PONE-D-20-31979R1 

Need for self-medication using over-the-counter psychoactive agents: A national survey in Japan 

Dear Dr. Shiina:

I'm pleased to inform you that your manuscript has been deemed suitable for publication in PLOS ONE. Congratulations! Your manuscript is now with our production department. 

Kind regards, 

on behalf of

Prof Kazutaka Ikeda 

Academic Editor

PLOS ONE